# SARS-CoV-2 antibody trajectories after a single COVID-19 vaccination with and without prior infection

Jia Wei[1,2], Philippa C. Matthews [1,3,4], Nicole Stoesser [1,5,6,7], Ian Diamond[8], Ruth Studley[8], Emma Rourke[8], Duncan Cook[8], John I. Bell[9], John N. Newton[10], Jeremy Farrar[11], Alison Howarth[1,5], Brian D. Marsden[1,12], Sarah Hoosdally[1], E. Yvonne Jones[1], David I. Stuart[1], Derrick W. Crook[1,5,6,7], Tim E. A. Peto[1,5,6,7], A. Sarah Walker[1,2,6,13,25], David W. Eyre [2,5,6,7,25], Koen B. Pouwels [6,14,25 ✉] & The COVID-19 Infection Survey team*

Given high SARS-CoV-2 incidence, coupled with slow and inequitable vaccine roll-out in many settings, there is a need for evidence to underpin optimum vaccine deployment, aiming to maximise global population immunity. We evaluate whether a single vaccination in individuals who have already been infected with SARS-CoV-2 generates similar initial and subsequent antibody responses to two vaccinations in those without prior infection. We compared anti-spike IgG antibody responses after a single vaccination with ChAdOx1, BNT162b2, or mRNA-1273 SARS-CoV-2 vaccines in the COVID-19 Infection Survey in the UK general population. In 100,849 adults median (50 (IQR: 37–63) years) receiving at least one vaccination, 13,404 (13.3%) had serological/PCR evidence of prior infection. Prior infection significantly boosted antibody responses, producing higher peak levels and/or longer half-lives after one dose of all three vaccines than those without prior infection receiving one or two vaccinations. In those with prior infection, the median time above the positivity threshold was >1 year after the first vaccination. Single-dose vaccination targeted to those previously infected may provide at least as good protection to two-dose vaccination among those without previous infection.

[1] Nuffield Department of Medicine, University of Oxford, Oxford, UK. [2] Big Data Institute, Nuffield Department of Population Health, University of Oxford, Oxford, UK. [3] The Francis Crick Institute, 1 Midland Road, London, UK. [4] Division of infection and immunity, University College London, London, UK. [5] Department of Infectious Diseases and Microbiology, Oxford University Hospitals NHS Foundation Trust, John Radcliffe Hospital, Oxford, UK. [6] The National Institute for Health Research Health Protection Research Unit in Healthcare Associated Infections and Antimicrobial Resistance at the University of Oxford, Oxford, UK. [7] The National Institute for Health Research Oxford Biomedical Research Centre, University of Oxford, Oxford, UK. [8] Office for National Statistics, Newport, UK. [9] Office of the Regius Professor of Medicine, University of Oxford, Oxford, UK. [10] European Centre for Environment and Human Health, University of Exeter, Truro, UK. [11] Wellcome Trust, London, UK. [12] Nuffield Department of Orthopaedics, Rheumatology and Musculoskeletal Sciences, University of Oxford, Oxford, UK. [13] MRC Clinical Trials Unit at UCL, UCL, London, UK. [14] Health Economics Research Centre, Nuffield Department of Population Health, University of Oxford, Oxford, UK. [25] These authors contributed equally: A. Sarah Walker, David W. Eyre, Koen B. Pouwels. *A list of authors and their affiliations appears at the end of the paper. ✉email: koen.pouwels@ndph.ox.ac.uk

COVID-19 vaccines have moderate to high efficacy in preventing infections, severe illness, hospitalisation, and death, including the mRNA vaccines BNT162b2 and mRNA-1273, and adenovirus vaccine ChAdOx1[1–3]. By March 2022, more than 11 billion doses had been administered globally[4]. However, 37% of the world's population remain unvaccinated, with low vaccination rates in many low-income countries[5,6]. Previous infection also confers protection against re-infection[7,8]. However, prior infection rates vary widely, with high seroprevalence estimates in South Africa (48.5%), Ecuador (44.8%) and Peru (43.5%) versus 0.71% in Australia and New Zealand at various stages of the pandemic[9].

Optimising global immunity and protection against infection is a priority, to minimise deaths, morbidity, and socio-economic losses. Major initiatives seek to address inequalities in vaccine availability, including the COVAX programme[10], but marked variation in global access persists. The WHO roadmap for prioritising COVID-19 vaccine use is focused on scale-up of equitable vaccine delivery[11]. Modelling studies suggest that prioritisation based on seropositivity substantially improves efficiency where seroprevalence is high[12]. Given this, a single-dose vaccine strategy for individuals with prior infection has been adopted in some settings during the pandemic (e.g., Netherlands[13], France, Italy, Germany[14]).

Understanding how prior infection influences antibody responses to vaccinations could help inform strategies to optimise population immunity. If a single vaccination invokes effective protection following prior infection, changing vaccine prioritisation as an interim measure may deliver higher population-level immunity faster, and make vaccination programmes more affordable.

Existing studies have focused on how prior infection affects initial peak antibody responses[15] or responses after two vaccinations[16–18], showing prior infection significantly boosts vaccine-mediated antibody levels[19–22]. However, the durability of antibody response after a single vaccination is unclear; in particular, whether a single dose can provide sustained protection for individuals with prior SARS-CoV-2 infection.

We used data from the United Kingdom's national COVID-19 Infection Survey (ISRCTN21086382), to investigate the relationship between prior infection and anti-trimeric spike IgG antibody responses following a single dose of ChAdOx1, BNT162b2, or mRNA-1273 vaccine. We compared the estimated duration of protection in those who had already been infected with SARS-CoV-2 before receiving a single vaccination to (i) those receiving a single vaccination without previous infection, and (ii) our previous findings for those receiving two vaccinations with and without previous infection[16]. Participants contributed predominantly monthly blood samples for antibody testing, which were scheduled independently of any vaccines received.

## Results and discussion

We included participants with at least one antibody measurement from 91 days before the first vaccination onwards up to the second vaccination (if received) or breakthrough infection post-first dose. From 8th December 2020 to 31st January 2022, 80,353 included participants received at least one dose of ChAdOx1 (10,207 [12.7%] with prior infection before the first vaccination), 57,181 at least one dose of BNT162b2 (10,116 [17.7%] with prior infection, reflecting many SARS-CoV-2 exposed healthcare workers receiving BNT162b2 early in the vaccination programme), and 4398 at least one dose of mRNA-1273 (807 [18.3%] with prior infection, reflecting this vaccine being used later in the pandemic in younger age groups) (Supplementary Table 1). The median age was 50 years (interquartile range IQR: 37–63). 75,593

(53.3%) were female, 131,365 (92.6%) reported white ethnicity, 1243 (0.9%) black ethnicity, 5699 (4.0%) Asian ethnicity, 2241 (1.6%) mixed ethnicity, and 1384 (1.0%) another ethnicity. 2577 (1.8%) were healthcare workers, and 33,599 (23.7%) reported having a long-term health condition.

We modelled antibody trajectories using measurements from 28 days post-first dose for all participants (i.e., the time of approximate peak levels in observed and modelled data, Supplementary Figs. 1, 2), accounting for the assay upper limit of quantification using interval censored models and adjusting for age, sex, ethnicity, reporting a long-term health condition or working in healthcare, and deprivation in order to directly estimate peak levels and antibody declines by these characteristics in contrast to previous descriptive analyses[16]. We excluded participants who did not mount an anti-S antibody response to first vaccination (defined as all antibody measurements <16 BAU/mL, including ≥1 measurement ≥21 days after the first dose [similar to previously[15,16]]): 4488 (7.4%), 1450 (4.2%), and 17 (0.7%) participants without prior infection, and 149 (1.9%), 52 (0.8%), 0 (0%) with prior infection receiving ChAdOx1, BNT162b2, and mRNA-1273, respectively.

59,101 participants with a single ChAdOx1 vaccination (7191 [12.2%] previously infected) contributed 73,963 antibody measurements ≥28 days post-first dose, median (IQR) [range] 2 (1-2) [1-3] measurements per participant. Assuming antibody levels declined exponentially based on previous descriptive analyses[16], using multivariable Bayesian linear mixed models (and accounting for censoring arising from the upper limit of quantification of the antibody assay) we estimated a median peak anti-spike IgG level of 447 BAU/mL (95% credible interval, CrI 432–462), and a median half-life of 85 days (95%CrI 76–97) for those who had already been infected with SARS-CoV-2 before one vaccination in the reference category (50 years, female, white ethnicity, not reporting a long-term health condition, not a healthcare worker, and deprivation percentile = 60). This peak was substantially higher than our previous estimates after a second ChAdOx1 vaccination in participants without prior infection (160 BAU/mL [157–162] for the same reference category), and the half-life was similar (81 days [79–83])[16]. In this study, those receiving one ChAdOx1 vaccination without prior infection had significantly lower peak levels, 84 BAU/mL (81–85), but a similar antibody half-life to those with prior infection, 93 days (89–99) (Fig. 1a, Supplementary Table 2). Supplementary Table 3 compares peak levels and half-lives for different ages, sex, and ethnicity. Previously infected participants reporting non-white ethnicity had higher post-first dose peak levels but slightly shorter half-lives regardless of prior infection (Table 1, Supplementary Table 2).

35,638 participants with a single BNT162b2 vaccination (5523 [15.5%] previously infected) contributed 43,163 antibody measurements ≥28 days post-first dose, median (IQR) [range] 1 (1-2) [1-3] measurements per participant. For those who had already been infected with SARS-CoV-2 before one vaccination, the estimated median peak antibody level was 432 BAU/mL (95% CrI 416–449), and the half-life was 266 days (175–544) at the reference category. The peak levels were lower than previously estimated following two BNT162b2 vaccinations without prior infection (974 BAU/mL [942–1009]), but half-life was substantially longer (51 days [50–53])[16]. In this study, those with one vaccination but no prior infection had lower peak levels (168 BAU/mL [165–172]) and shorter half-lives (46 days [44–47]) (Fig. 1b, Supplementary Table 2). Similar to ChAdOx1, participants reporting non-white ethnicity had higher peak levels (Table 1, Supplementary Table 2).

2597 participants with one mRNA-1273 vaccination (414 [15.9%] previously infected) contributed 2975 antibody measurements ≥21 days post-first dose, median (IQR) [range] 1 (1-2)

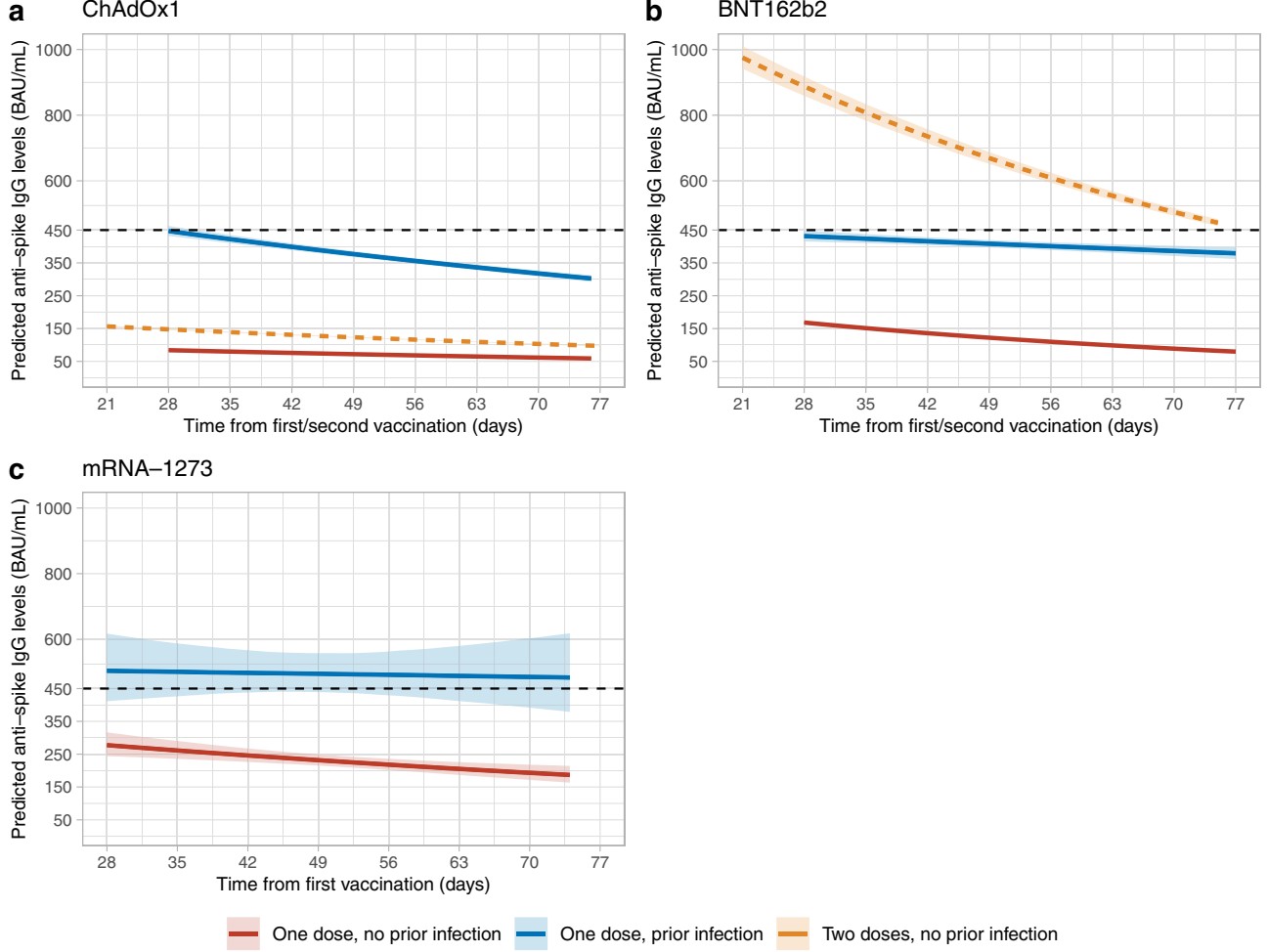

**Fig. 1 Posterior predicted trajectories (95% credible interval) of anti-spike IgG levels from 28 days post-first dose by prior infection status. a** 59,101 participants who received at least a single ChAdOx1 vaccination. **b** 35,638 participants who received at least a single BNT162b2 vaccination. **c** 2597 participants who received at least a single mRNA-1273 vaccination. Plotted at reference categories: 50 years, female, white ethnicity, not reporting a long-term health condition, not a healthcare worker, and deprivation percentile = 60. Black dotted line shows the upper quantification limit of 450 BAU/mL. Orange dotted lines in panel a and b were predicted trajectories starting from 21 days post-second dose for 92,584 and 51,034 participants who received two ChAdOx1 and BNT162b2 vaccination without prior infection reproduced from our previous analysis, plotted at the same reference categories[16]. m1273-RNA was not included in this previous analysis due to insufficient data at the time.

[1-2] measurements per participant. For those who had already been infected with SARS-CoV-2 before one vaccination the estimated peak level was 505 BAU/mL (413–619) at the reference category, and the half-life was 768 days (75-Not estimable). We were not able to estimate peak and half-life after two vaccinations in our previous study; in this study participants without prior infection had lower peak levels after one vaccination (278 BAU/mL (245–317)) and half-life was estimated to be 80 days (51–178) (Table 1, Fig. 1c, Supplementary Table 2). mRNA-1273 elicited higher antibody levels (especially in those without prior infection) and had a lower percentage of seronegative non-responders versus BNT162b2 (0.7% vs 4.2%), consistent with findings after two vaccinations among US healthcare workers[23].

Given the strong effects of different factors on response to one vaccination in multivariable models (Supplementary Table 2), we estimated the time from first vaccination to levels falling to the antibody positivity threshold (23 BAU/mL, see Methods) in different subgroups. For those with prior infection, for 20-year-old following one vaccination the estimated median durations were 340–440 days for those with ChAdOx1, >680 days for those with BNT162b2, and 300–900 days with mRNA-1273. For 40–60-year-olds with mRNA-1273, antibody levels in some groups were not

estimated to decline and the upper credible intervals could not be defined (Fig. 2). Conditional on seroconverting after one dose as described above, older participants had longer durations of seropositivity than younger participants following one dose of any of the three vaccines due to longer half-lives despite lower peak levels (Supplementary Table 2). Females and those without long-term health conditions also had longer estimated durations of seropositivity following one dose of any of the three vaccines. In our previous analysis of responses post-second vaccination[16], the time from the second dose to antibody levels falling to 23 BAU/mL in those without prior infection was estimated to be around 250 and 270–330 days for those receiving two ChAdOx1 or BNT162b2 doses[16] respectively, much shorter than with previous infection and one vaccination in this analysis, especially in the older ages.

To provide context to the observed antibody levels in this study, we previously found that levels of 107 and 94 BAU/mL were associated with 67% protection against new Delta infection in those without prior infection but vaccinated with ChAdOx1 and BNT162b2, respectively, compared with 33 BAU/mL in those unvaccinated with prior infection[16]. Equivalent levels could not be estimated for those with prior infection and vaccinated due to

**Table 1 Posterior predicted median peak levels (BAU/mL) and half-lives (days) with 95% credible intervals in participants received one vaccination with prior infection, two vaccinations without prior infection, and one vaccination without prior infection, by vaccine type.**

| Vaccine | Age | Group | Peak level (BAU/mL) | | | Half-life (days) | | |
|---|---|---|---|---|---|---|---|---|
| | | | One dose with prior infection | Two doses without prior infection (from ref. 16) | One dose without prior infection | One dose with prior infection | Two doses without prior infection (from ref. 16) | One dose without prior infection |
| ChAdOx1 | 50 | White female | 447 (432–462) | 160 (157–162) | 84 (83–85) | 85 (76–97) | 81 (79–83) | 93 (89–99) |
| | 50 | White male | 416 (402–431) | 154 (152–157) | 78 (77–80) | 77 (70–87) | 81 (79–83) | 84 (80–89) |
| | 50 | Non-white females | 637 (607–670) | 201 (195–207) | 120 (115–125) | 75 (65–88) | 73 (70–76) | 81 (72–92) |
| | 50 | Non-white males | 594 (565–625) | 195 (189–201) | 112 (107–116) | 69 (60–80) | 73 (70–77) | 74 (66–83) |
| BNT162b2 | 50 | White female | 432 (416–449) | 974 (942–1009) | 168 (165–172) | 266 (175–544) | 52 (50–53) | 46 (44–47) |
| | 50 | White male | 397 (382–413) | 836 (807–864) | 155 (152–158) | 256 (168–500) | 51 (50–53) | 45 (43–47) |
| | 50 | Non-white females | 517 (487–549) | 1114 (1045–1190) | 202 (192–212) | 204 (127–505) | 51 (48–54) | 43 (39–48) |
| | 50 | Non-white males | 476 (448–506) | 955 (894–1019) | 186 (176–195) | 198 (124–470) | 51 (48–54) | 43 (39–48) |
| mRNA-1273 | 50 | White female | 505 (413–619) | | 278 (245–317) | 768 (75–Not estimable) | | 80 (51–178) |
| | 50 | White male | 447 (365–546) | | 246 (218–278) | 169 (57–Not estimable) | | 59 (43–97) |
| | 50 | Non-white females | 462 (348–615) | | 255 (200–325) | Not estimable (86–Not estimable) | | 184 (50–Not estimable) |
| | 50 | Non-white males | 409 (306–544) | | 225 (176–288) | Not estimable (62–Not estimable) | | 100 (41–Not estimable) |

Estimates for two vaccinations without prior infection were based on our previous analysis[16]. All estimates are at the reference age (50-year-old) and separated by sex (female vs male) and ethnicity (white vs non-white).

insufficient data[16]. However, since levels associated with the same degree of protection were lower for unvaccinated but previously infected individuals, if we conservatively assume the threshold levels are similar post any vaccination, in those with prior infection the duration providing >67% protection is estimated to be around 170–220 days for a single ChAdOX1 vaccination, and over a year for a single BNT162b2 vaccination. Since the duration providing >67% protection in unvaccinated individuals with natural infection was estimated to be 1–2 years in our previous analysis[16], it is highly likely that the duration of protection is >1 year for those with prior infection receiving a single vaccination. Using similar assumptions for those with previous infection, 2 vaccinations of would result in an estimated duration of 67% protection of 180–200 for ChAdOX1 and 240–300 days for BNT162b2, respectively[16]. Therefore, for people with prior infection, two vaccinations did not provide longer protection than a single vaccination. In those without prior infection, using these thresholds, a single ChAdOx1 vaccination would not reach the required antibody level, while a single BNT162b2 vaccination would provide 50–100 days of protection for people <60 years (Supplementary Fig. 3).

Higher antibody levels post first SARS-CoV-2 vaccination in previously infected individuals have been reported[19–22], but none of these studies, nor our previous descriptive analysis of response to first vaccinations[16], estimated the trajectory of antibody response post first vaccination, overall or according to important characteristics such as age. We found that in those with prior infection, not only were antibody peak levels higher for all three vaccines compared to those without prior infection, by 200–400 BAU/mL, showing a substantial boosting effect of prior infection, but the subsequent waning was also slower following BNT162b2 and mRNA-1273, supporting sustained protection from a single dose in previously infected individuals. The combination of prior infection with a single vaccination resulted in similar antibody levels regardless of vaccine type (Fig. 1), despite single ChAdOx1 vaccination resulting in lower peak levels than BNT162b2 in those without prior infection. The higher peak levels from mRNA-1273 than BNT162b2 could be potentially explained by a higher spike protein delivery in mRNA-1273.

Our results provide data which could be used to compare different approaches to global immunisation strategies, deploying limited resources in the most effective way to deliver maximum population immunity rapidly during a period when incidence remains high and vaccine access is not yet universal. Given the low percentage of vaccinated individuals and high SARS-CoV-2 seroprevalence in some settings[24–26], assuming widespread prior infection, a single vaccination could provide population-level protection as a short-term measure.

However, such a blanket approach would result in infection-naive individuals initially receiving one vaccination only, which provides suboptimal protection[8]. To reduce this risk, where previous infection is expected to be high, another option would be to stratify individual vaccination based on an affordable and rapid lateral flow immunoassay (LFIA) for antibody detection, and information on previous PCR positivity[27–31]. An extensive comparison of different fingerprick-based LFIA antibody tests reported high specificity (98.8–99.8%)[32], meaning that there would be few false positives in a high prevalence setting. Sensitivity was lower (69–86%)[32], which would reduce efficiency, but not vaccine programme effectiveness. Although exact prices will vary by manufacturer and may be negotiated, the cost can be as low as US$1–2 per test[33]. In comparison, a ChAdOx1 vaccine costs around $4 per dose excluding delivery and storage costs[34].

For an unvaccinated population of 10 million individuals willing to be vaccinated with a true seroprevalence of 50%, a LFIA with 99% specificity and 80% sensitivity costing $1.5 per test, and

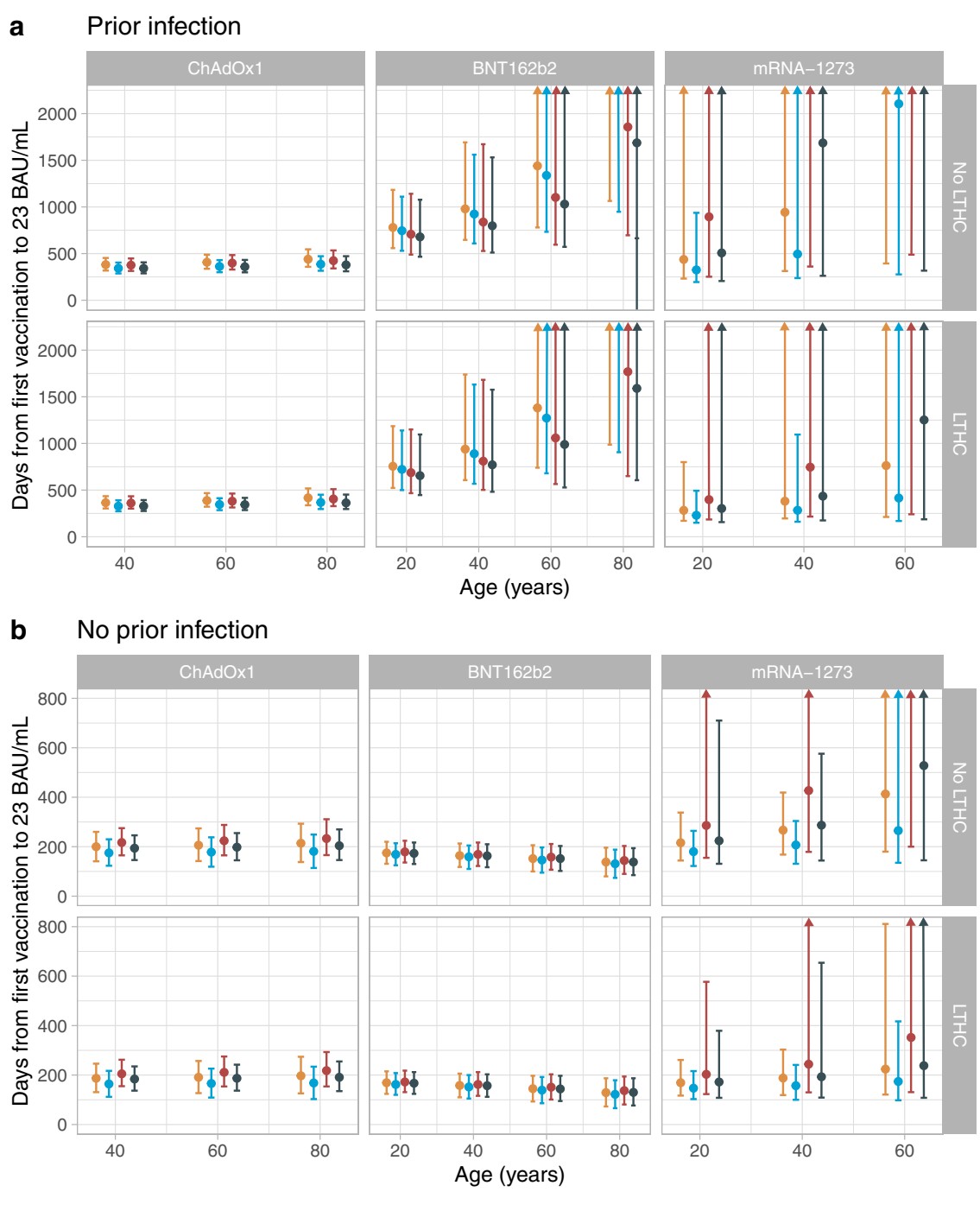

**Fig. 2 Posterior predicted mean days (95% credible interval) from the first vaccination to the positivity threshold of 23 BAU/mL. a** In those with evidence of prior infection (n = 7191, 5523, 414 for ChAdOx1, BNT162b2, mRNA-1273, respectively). **b** in those without evidence of prior infection (n = 51,910, 30,115, 2183 for ChAdOx1, BNT162b2, mRNA-1273, respectively). Estimates were separated by age (predicted at 20, 40, 60, and 80-year-old), sex, ethnicity, long-term health condition (LTHC), and vaccine type. The y-axis is truncated at 2000 days (panel **a**) for visualisation. For ChAdOx1, the 20-year-old group is not plotted because the vast majority of those receiving ChAdOx1 were ≥40 years. For mRNA-1273, the 80-year-old is not plotted because the vast majority of those receiving mRNA-1273 were ≤60 years. Equivalent estimates after second vaccination are provided in ref. [16].

a cost of $5 per vaccination (including delivery costs and storage), an LFIA test performed at the first vaccination visit would correctly identify 4,050,000 individuals as having antibodies and eligible for a one-dose schedule – of which 50,000 would be incorrectly identified as having antibodies – the remaining 5,950,000 being invited for a second vaccination according to the agreed dosing interval. This would result in 4.05 million fewer vaccinations needed in the short-term to produce equivalent population immunity to a universal two-dose campaign. This approach delivers a cost saving of $5.25 million which could be re-invested in securing robust long-term vaccine access (for results with other settings regarding LFIA test sensitivity and

specificity, population size, true antibody prevalence, costs per vaccination and LFIA, see https://herc.shinyapps.io/Serology_vaccine_prioritisation/). While our study provides some support for modelling studies that suggest that prioritisation based on seropositivity may substantially improve efficiency of vaccine allocation where seroprevalence is high, basing vaccination policies on studies combining antibody trajectories with correlates of protection against infection and modelling studies alone is challenging.

Study strengths include our use of fully adjusted Bayesian linear mixed interval-censored models, which account for antibody measurements above the limit of quantification (450 BAU/mL), to directly estimate changes in antibody levels after the first vaccination adjusting for multiple characteristics in a younger population than our previous descriptive study of first-vaccination responses[16], our ability to include the m1273-RNA vaccine in this analysis, and compare with both one and two vaccinations without previous infection in the general population. Simulation studies demonstrated that our models were robust to the assay limit and could produce accurate estimates of antibody peak levels and half-lives (simulations including interval censoring at similar levels to our dataset showed good coverage and no/negligible bias, Supplementary Table 4). Study limitations include that we did not measure other immune responses, including T cell or innate immune responses. We only measured anti-spike IgG antibody levels using a single assay; however, we calibrated antibody levels to WHO BAU/mL units for comparison with other studies. Anti-spike IgG antibody levels are only an indirect proxy for vaccine efficacy; however, assay results have been previously shown to correlate closely with neutralising activity and vaccine protection (correlates of protection), thus could be used to inform vaccine policies[16]. In those with prior infection, older participants had slower antibody declines, which could be due to the study design, where only those who seroconverted were included, as only these participants could have their antibody declines analysed. Older participants who seroconverted may have had more sustained immune responses than younger participants overall. We did not have enough data to model antibody response in older participants who received mRNA-1273, as 98% of participants who received mRNA-1273 were <50 years. Most of our participants reported white ethnicity (92.6%), so wider generalisability to non-white ethnic groups is less well defined and our data were insufficient to model other ethnic groups separately. However, estimated durations of protection for non-white participants were broadly as long or longer than for white participants, with non-white ethnicity associated with higher peak levels after a single ChAdOX1 or BNT162b2 vaccination.

Data are still awaited to determine the impact of the Omicron variant, and the relationship between prior infection/vaccination and immunological protection from this and other new variants. Given this, all approaches to vaccine scheduling will need to remain under intense scrutiny, and the international focus must remain firmly on assuring equitable access to full vaccination in all population settings, which will also mitigate the potential for further variants of concern to emerge.

In summary, prior infection significantly boosts antibody responses after a single ChAdOx1, BNT162b2, or mRNA-1273 vaccination, producing higher peak levels and/or longer half-lives, comparable or even better to those obtained from two vaccinations in those without prior infection. Based on the positivity threshold and previously reported correlates of protection, those with prior infection could be protected from infection for >1 year after a single vaccination. While recent studies show that a third vaccination boosts antibody responses[35] and provides better protection against infection than two vaccinations[36], and two vaccinations plus prior infection provides better protection

against re-infection than prior infection alone[8], a large part of the global population has not yet had the chance to get their first vaccination due to limited vaccine supplies. These results could stimulate further studies assessing the impact of a one-dose strategy for those with previous infection, together potentially providing an evidence-base to optimise population immunity in the context of resource limitations, while international SARS-CoV-2 immunisation programmes are scaled up and secured.

## Methods

**Population and survey.** The UK's Office for National Statistics (ONS) COVID-19 Infection Survey (CIS) (ISRCTN21086382) randomly and continuously recruited private households to provide a representative sample across its four countries (England, Wales, Northern Ireland, and Scotland). At the first visit, participants were asked for consent for optional follow-up visits every week for the next month, then monthly for 12 months or to April 2022. Written informed consent was taken from individuals ≥2 years (children aged <2 years were not eligible for the study). For those 2–15 years this consent was obtained from parents/carers, while those 10–15 years also provided written assent. Participants ≥16 years gave their own consent.

Socio-demographic characteristics, behaviours, and vaccination data were collected. Combined nose and throat swabs were taken from all consenting participants for SARS-CoV-2 PCR testing. Blood samples were taken from individuals ≥16 years from 10-20% randomly selected households monthly for serological testing, and participants who tested swab positive and their household members were also invited to provide blood samples at follow-up visits. Details on the sampling design are provided elsewhere[37]. From April 2021, additional participants were invited to provide blood samples monthly to assess vaccine responses. The study protocol is available at https://www.ndm.ox.ac.uk/covid-19/covid-19-infection-survey/protocol-and-information-sheets. The study received ethical approval from the South Central Berkshire B Research Ethics Committee (20/SC/0195).

**Vaccination data.** Self-reported vaccinations were obtained from participants at visits, including vaccination type, number of doses, and vaccination dates. Participants from England were also linked to the National Immunisation Management Service (NIMS), which contains all individuals' vaccination data in the English National Health Service COVID-19 vaccination programme. There was good agreement between self-reported and administrative vaccination data (98% on type and 95% on date[38]). We used vaccination data from NIMS where available for participants from England, and otherwise data from the survey.

**Laboratory testing.** Combined nose and throat swabs were tested at high-throughput national "Lighthouse" laboratories in Glasgow (from 16 August 2020 to present) and Milton Keynes (from 26 April 2020 to 8 February 2021). PCR outputs were analysed using UgenTec Fast Finder 3.300.5 (TaqMan 2019-nCoV Assay Kit V2 UK NHS ABI 7500 v2.1; UgenTec), with an assay-specific algorithm and decision mechanism that allows conversion of amplification assay raw data into test results with minimal manual intervention. Positive samples are defined as having at least a single N and/or ORF1ab gene detected, and PCR traces exhibited an appropriate morphology. The S gene alone is not considered as positive[37].

SARS-CoV-2 antibody levels were tested on venous or capillary blood samples using an ELISA detecting anti-trimeric spike IgG developed by the University of Oxford[37,39]. Normalised results are reported in ng/ml of mAb45 monoclonal antibody equivalents. Before 26 February 2021, the assay used fluorescence detection as previously described, with a positivity threshold of 8 million units validated on banks of known SARS-CoV-2 positive and negative samples[39]. After this, it used a commercialised CE-marked version of the assay, the Thermo Fisher OmniPATH 384 Combi SARS-CoV-2 IgG ELISA (Thermo Fisher Scientific), with the same antigen and colorimetric detection. mAb45 is the manufacturer-provided monoclonal antibody calibrant for this quantitative assay. To allow conversion of fluorometrically determined values in arbitrary units, we compared 3840 samples which were run in parallel on both systems. A piece-wise linear regression was used to generate the following conversion formula:

$$
\begin{aligned}
log_{10}(mAb45\ units) = {}& 0.221738 + 1.751889e\text{-}07 * fluorescence\_units \\
& + 5.416675e\text{-}07 * (fluorescence\_units > 9190310) \\
& * (fluorescence\_units\text{-}9190310)
\end{aligned} \tag{1}
$$

We calibrated the results of the Thermo Fisher OmniPATH assay into WHO international units (binding antibody unit, BAU/mL) using serial dilutions of National Institute for Biological Standards and Control (NIBSC) Working Standard 21/234. The NIBSC 21/234 Working Standard has been previously calibrated against the WHO International Standard for anti-SARS-CoV-2 immunoglobulin (NIBSC code 20/136), with anti-spike IgG potency of 832 BAU/mL (95%CI 746-929). We generated 2-fold dilutions of 21/234 between 1:400 and 1:8000 from three separate batches on three separate days. Results from a total of 63 diluted samples were merged and a linear regression model fitted constrained to

have an intercept of zero to convert mAB45 units in ng/ml for samples diluted at 1:50 to BAU/mL:

$$BAU/mL = 0.559 * [\text{mAb45 concentration in ng/mL at } 1:50]$$

23 BAU/mL was used as the threshold for an IgG positive or negative result (corresponding to the 8 million units with fluorescence detection). Given the lower and upper limits of the assay, measurements <1 BAU/mL (2545 observations, 0.8%) and >450 BAU/mL (29,035 observations, 8.7%) were truncated at 1 and 450 BAU/mL, respectively.

**Statistical analysis**. For the current study, participants aged ≥16 years who received at least a single vaccination with ChAdOx1 or BNT162b2 or mRNA-1273 with antibody measurements from 8th December 2020 until 31st January 2022 were included. Participants with prior infection (before vaccination) were defined as (1) having a positive PCR swab test in the survey or the linked English national testing programme; (2) having a positive anti-spike IgG result (≥23 BAU/mL) before vaccination; (3) having two consecutive positive anti-nucleocapsid IgG results (≥17 BAU/mL); or (4) self-reporting a positive swab test in the survey. The infection date was defined as the earliest recorded date from the above definitions. Age was truncated at 85 years to reduce the influence of outliers.

To estimate antibody waning, we excluded a small number of participants who were considered as non-responders after the first dose, defined as all antibody measurements being <16 BAU/mL and having at least one antibody measurement 21 days after the first dose ($N = 4488$ excluded for ChAdOx1, $N = 1450$ excluded for BNT162b2, $N = 17$ excluded for mRNA-1273).

Bayesian linear mixed interval-censored models were used to estimate changes in antibody levels after the first vaccination with ChAdOX1, BNT162b2, or mRNA-1273. Antibody measurements taken after the second vaccination or after infection that happened post-first dose were excluded. We included antibody measurements from 28 days post-first dose to reflect the peak level. We excluded measurements taken after the 90th percentile of the observed time points to avoid outlier influence (76, 77, and 74 days post-first dose for ChAdOx1, BNT162b2, and mRNA-1273).

We used a multivariable model to examine the association between peak levels and antibody half-lives with continuous age (16–85 years, predicted values at 20-, 40-, 60-, and 80-year-olds to represent young, middle-aged, and older adults), sex, ethnicity (white vs non-white), reporting having a long-term health condition, reporting working in patient-facing healthcare, deprivation percentile, and prior infection status. Non-white ethnicities were combined because of small sample sizes. We assumed an exponential fall in antibody levels over time, i.e., a linear decline on a log2 scale[16]. To examine non-linearity in antibody declines, especially the potential for the rate of antibody decline to flatten, we additionally fitted a model using 4-knot splines for time (knots placed at 10th, 33rd, 67th, and 90th of observed time points) and compared the model fit with the log-linear model using the leave-one-out cross-validation information criterion (LOOIC). We found that the spline model had a higher LOOIC (indicating a worse model fit) than the log-linear model for ChAdOx1 (155,331 vs 138,883) and mRNA-1273 (7586 vs 7570). The model fit for BNT162b2 was slightly better in the spline model than the log-linear model (107,858 vs 108,863), however, for all three models the estimated trajectories were similar and there was no evidence of antibody decline flattening up to 77 days post-first vaccination (Supplementary Fig. 4), so we used the log-linear model for the analysis.

Population-level fixed effects, individual-level random effects for intercept and slope, and correlation between random effects were included in the models. The outcome was right-censored at 450 BAU/mL reflecting truncation of IgG values at the upper limit of quantification (i.e., all measurements truncated to 450 BAU/mL were considered to be >450 BAU/mL in analyses). For each model, weakly informative priors were used. Four chains were run per model with 4000 iterations and a warm-up period of 2000 iterations to ensure convergence, which was confirmed visually and by ensuring the Gelman-Rubin statistic was <1.05. 95% credible intervals were calculated using highest posterior density intervals.

To evaluate whether the censored Bayesian linear model could recover the true intercept and slope, and correlation between them, in the presence of similar percentage of observations being above the upper limit of quantification as observed in our study, we performed a simulation study. The correlation between intercept and slope were varied across scenarios between −0.5 and 0.5, and slopes and intercepts were chosen to be similar to those observed in the real-world data. Within each scenario, we chose two parameter sets to line up with the different proportion of censored measurements (the proportion of censored data is 11% for ChAdOx1, 19% for BNT162b2, and 38% for mRNA-1273). We used 1000 as the sample size and performed 100 repetitions of each simulation, to reduce computational burden, providing conservative estimates of model performance in terms of bias and coverage.

Data cleaning and preparation were performed in STATA MP 16. All analyses were performed in R 4.1 using the following packages: tidyverse (version 1.3.1), brms (version 2.15.0), arsenal (version 3.4.0), cowplot (version 1.1.1), bayesplot (version 1.8.1), and tidybayes (version 3.0.1).

**Reporting summary**. Further information on research design is available in the Nature Research Reporting Summary linked to this article.

## Data availability
Data are still being collected for the COVID-19 Infection Survey. De-identified study data are available for access by accredited researchers in the ONS Secure Research Service (SRS) for accredited research purpose under part 5, chapter 5 of the Digital Economy Act 2017. Individuals can apply to be an accredited researcher using the short form on https://researchaccreditationservice.ons.gov.uk/ons/ONS_registration.ofml. Accreditation requires completion of a short free course on accessing the SRS. To request access to data in the SRS, researchers must submit a research project application for accreditation in the Research Accreditation Service (RAS). Research project applications are considered by the project team and the Research Accreditation Panel (RAP) established by the UK Statistics Authority at regular meetings. Project application example guidance and an exemplar of a research project application are available. A complete record of accredited researchers and their projects is published on the UK Statistics Authority website to ensure transparency of access to research data. For further information about accreditation, contact Research.Support@ons.gov.uk or visit the SRS website. Source data are provided with this paper.

## Code availability
A copy of the analysis code is available at https://github.com/jiaweioxford/COVID19_antibody_response_first_dose. https://doi.org/10.5281/zenodo.6575007

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

## Acknowledgements

We are grateful for the support of all COVID-19 Infection Survey participants. This study is funded by the Department of Health and Social Care with in-kind support from the Welsh Government, the Department of Health on behalf of the Northern Ireland Government and the Scottish Government. JW is supported by University of Oxford and the China Scholarship Council. A.S.W., T.E.A.P., N.S., D.E., K.B.P. are supported by the National Institute for Health Research (NIHR) Health Protection Research Unit in Healthcare Associated Infections and Antimicrobial Resistance (NIHR200915), a partnership between the UK Health Security Agency (UKHSA) and the University of Oxford. A.S.W. and T.E.A.P. are also supported by the NIHR Oxford Biomedical Research Centre. K.B.P. is also supported by the Huo Family Foundation. A.S.W. is also supported by core support from the Medical Research Council UK to the MRC Clinical Trials Unit [MC_UU_12023/22] and is an NIHR Senior Investigator. P.C.M. is funded by Wellcome (intermediate fellowship, grant ref 110110/Z/15/Z) and holds an NIHR Oxford BRC Senior Fellowship award. D.W.E. is supported by a Robertson Fellowship and an NIHR Oxford BRC Senior Fellowship. N.S. is an Oxford Martin Fellow and holds an NIHR Oxford BRC Senior Fellowship. The views expressed are those of the authors and not necessarily those of the National Health Service, NIHR, Department of Health and Social Care, or UKHSA.

## Author contributions

The study was designed and planned by A.S.W., J.F., J.B., J.N., I.D. and K.B.P., and is being conducted by A.S.W., R.S., D.C., E.R., A.H., B.M., T.E.A.P., P.C.M., N.S., S.H., E.Y.J., D.I.S., D.W.C. and D.W.E. This specific analysis was designed by J.W., D.W.E., A.S.W. and K.B.P. J.W. contributed to the statistical analysis of the survey data. J.W., D.W.E., K.B.P. and A.S.W. drafted the manuscript and all authors contributed to interpretation of the data and results and revised the manuscript. All authors approved the final version of the manuscript.

## Competing interests

D.W.E. declares lecture fees from Gilead, outside the submitted work. No other author has a conflict of interest to declare.

## Additional information

**The COVID-19 Infection Survey team**

Tina Thomas[8], Duncan Cook[8], Daniel Ayoubkhani[8], Russell Black[8], Antonio Felton[8], Megan Crees[8], Joel Jones[8], Lina Lloyd[8], Esther Sutherland[8], Emma Pritchard[1], Karina-Doris Vihta[1], George Doherty[1], James Kavanagh[1], Kevin K. Chau[1], Stephanie B. Hatch[1], Daniel Ebner[1], Lucas Martins Ferreira[1], Thomas Christott[1], Wanwisa Dejnirattisai[1], Juthathip Mongkolsapaya[1], Sarah Cameron[1], Phoebe Tamblin-Hopper[1], Magda Wolna[1], Rachael Brown[1], Richard Cornall[1], Gavin Screaton[1], Katrina Lythgoe[2], David Bonsall[2], Tanya Golubchik[2], Helen Fryer[2], Stuart Cox[15], Kevin Paddon[15], Tim James[15], Thomas House[16], Julie Robotham[17], Paul Birrell[17], Helena Jordan[18], Tim Sheppard[18], Graham Athey[18], Dan Moody[18], Leigh Curry[18], Pamela Brereton[18], Ian Jarvis[19], Anna Godsmark[19], George Morris[19], Bobby Mallick[19], Phil Eeles[19], Jodie Hay[20], Harper VanSteenhouse[20], Jessica Lee[21], Sean White[22], Tim Evans[22], Lisa Bloemberg[22], Katie Allison[23], Anouska Pandya[23], Sophie Davis[23], David I. Conway[24], Margaret MacLeod[24] & Chris Cunningham[24]

[15]Oxford University Hospitals NHS Foundation Trust, Oxford, UK. [16]University of Manchester, Manchester, UK. [17]Health Improvement Directorate, Public Health England, London, UK. [18]IQVIA, London, UK. [19]National Biocentre, Milton Keynes, UK. [20]Glasgow Lighthouse Laboratory, London, UK. [21]Department of Health and Social Care, London, UK. [22]Welsh Government, Cardiff, UK. [23]Scottish Government, Edinburgh, UK. [24]Public Health Scotland, Edinburgh, UK.

