## [Peer Review File · Nature Communications]

SARS-CoV-2 antibody trajectories after a single COVID-19 vaccination with and without prior infectionREVIEWER COMMENTS

Reviewer #1 (Remarks to the Author):

This article uses a large cohort of UK individuals that received multiple vaccine platforms for COVID-19. The authors measure antibody levels after a single or two-dose vaccine in individuals with or without prior SARS-CoV-2 infection. The authors then model antibody decay and perform comparisons. The constricted availability of vaccines make this observation important as it could affect vaccine policy. However, with the emergence of Omicron and other variants it is now more abundant than ever that multiple doses are required for protection and this strategy may not be viable. For example, there is significant Omicron re-infection in individuals with prior infection and significant infection even in vaccinated individuals. Below are specific comments that would strengthen the manuscript.

The major limitation was the timing of the blood sampling was not understood by this reviewer. Then the modeling of antibody decay did not reflect real-world data. First, in figure 1C, levels actually increase over time. Second, in figure 2, older individuals have longer durability. The authors should spend more time describing when the sample was taken and how the modeling of decay is performed.

A second major issue was the correlation to protection equivalence and conjecture about vaccine policy. This is modeled antibody decay, real data would need to be evaluated to inform a vaccine policy change. This would include defining correlates of protection.

1. Remove descriptive and personal inferences from abstract and text. Ie "slow and inequitable vaccine roll-out" Are there scientific studies of this to add. Would single doses be faster and more equitable? These statements do not add to the findings. Including the title.
2. Spike binding antibodies do not equate to vaccine protection or efficacy. This needs to be more clear and suggestions of changing vaccine policy due to antibody binding titers that have not been shown to be a correlate of immunity needs to be removed.
3. There have been studies that examine durability of antibody responses in those with prior infection and vaccination. (eg. Fraley et al. Clin. Infectious diseases 2021.; Zhong et al. JAMA 2021).
4. Comment on individuals that did not respond to first dose. This was over 6,000 people in this study. If you expanded that to global numbers that would be a lot of individuals without any type of immunity if only received a single dose.
5. More clarity on when the antibody levels are measured for each group should be provided.
6. It has been demonstrated with real world data that older individuals have lower durability so there must be an error in the modeling. (Figure 2).
7. The reference used for correlates of protection levels has not been peer reviewed. The logic of having lower levels does not track with protection. This results section is pure speculation with modeled data.

Reviewer #2 (Remarks to the Author):

Interesting study, for clinicians, policy makers, and the general scientific community.

Only major comments relate to revising the manuscript to make it more concise and to improve the verbiage - see below.

Title would be improved by revising to read: "...vaccine supplies: prior infection and anti-spike..."

The manuscript would benefit from editing to make it more concise, e.g., in abstract, fourth line

from the bottom, revise to read: "...threshold was > 1 year after the first dose."

The term "impact" is really not a scientific term. For example, in first paragraph of methods in the main manuscript, the second line would be improved by revising to read: "investigate the relationship between prior infection on..."

The term "at speed" is used at least twice in the manuscript - revise to be more clear.

Reviewer #3 (Remarks to the Author):

This is a timely manuscript given the increasing prevalence of seropositivity to SARS-CoV-2 in the population, and on-going discussions about vaccine policy and best use of limited supply. The duration findings are particularly relevant.

There are no page or line numbers included in the manuscript, so I will describe the location of questions or proposed edits as best I can below.

Main: Would update section on vaccination rates, and include a web reference rather than older, fixed date references for this informatin, which is in flux.

Main.

"and potentially explained by a higher spike protein delivery in MRNA-1273" should be removed or in the discussion - it isn't a result but rather an interpretation of a result.

Did you monitor infection after vaccination?

IN the paragraph that beings "we previously" on the 5th line, give the virus strain to which the protection and antibody levels are shown.

Discussion.

Missing from the discussion is mention of variants of concern, how these might affect results, interpretation and suggestions for programmatic use of number of doses? This seems to be a major deficiency in the current context.

Pre-testing at time of first vaccination for antibody seems impractical in most settings - perhaps it could be operationalized in certain high resource settings. How dependent is the antibody testing on training? What is the access to these tests?

Methods.

Was the decision to exclude non-responders pre-stated, or was that a posthoc decision? How many participants were in the category and how did it influence results?

Reviewer #4 (Remarks to the Author):

The authors present a study that uses antibody binding measurements in a large population level cohort study to determine duration of antibody positivity after vaccination in individuals with and without prior infection. They report that antibody positivity is longer in previously infected individuals who receive one vaccination than in individuals who receive one dose of a vaccine, but not compared to naïve individuals who receive two doses of an mRNA vaccine. The authors argue that this supports a strategy, in vaccine limited contexts, to screen and vaccinate with only one dose those who have had prior infection, but two doses for those without prior infection.

I have a number of major technical and conceptual concerns. In particular, that the primary results presented in this manuscript appear to be also reported in the authors other manuscript referenced as reference 16 in this paper.

Major concern over novelty:

1) This manuscript does not present much that is novel, or perhaps at best presents a very incremental improvement on the authors other paper (reference 16).

The authors here report two main findings, perhaps best summarised in the abstract as:

a. "Prior infection significantly boosted antibody responses for all three vaccines, producing a higher peak level and longer half-life, and a response comparable to those without prior infection receiving two vaccinations."

b. "In those with prior infection, median time above the positivity threshold was estimated to last for >1 year after the first dose."

However, the authors have another manuscript online, which they cite in this paper (reference 16), which reports the exact same conclusions from related data.

a. "Prior infection significantly increased antibody peak level and half-life with both vaccines."

b. "protection for vaccinated participants with prior infection, conservatively assuming threshold levels were similar to those vaccinated without prior infection and given their half-lives were longer, the duration of protection could last for >1 year."

I see that the current study has some extra mRNA-1273 data which was not present in the other paper, though this data does not greatly inform the manuscripts overall conclusions. The authors must make clear what is novel in this manuscript over their other manuscripts, but it is my feeling that this work is only an incremental advance on the authors reference 16.

Major technical concerns:

2) The estimate of half-life may be confounded by the Upper limit of quantification.

One of the potentially novel aspects of this paper over the authors other paper is the modelling of decay half-lives and peak antibody titers for different groups of individuals. However, this seems confounded by the upper limit of detection of the data on antibody titer used (450 BAU/ml). When fitting a model to estimate the peak level (intercept) and decay rate (slope), the two parameters can easily trade off against each other. That is, sometimes the model will have trouble determining if there was a high peak and rapid decay or a low peak and slow decay. The authors report a number of results where a lower peak was observed in one group but a longer half-life, especially.

"For those with prior infection, the estimated median peak antibody level was 434 BAU/mL (95%CrI 417-454), and the half-life was 333 days (193-1309) at the reference category. The peak levels were lower than previously reported following two BNT162b2 vaccinations without prior infection (974 BAU/mL [942-1009]), but half-life was substantially longer (51 days [50-53])¹⁶."

However, this is worrying because the Upper limit of quantification of the assay is 450. Even though the authors have attempted to deal with the limit of detection using censoring (which is commendable), it is hard to see how the results that the peak is just below the Upper LOD in one group and well above the Upper LOD in another group is going to be a robust result given the LOD is 450. Since the half-lives often trade off with estimated peak level its perhaps unsurprising that one has a long half-life and lower peak, and the other has a high peak and rapid decay. The authors need to demonstrate that their model can robustly estimate the peak level of antibody near or above the upper limit of detection.

The authors would have to do this with a model validation approach, to demonstrate their model is robust to the upper limit of detection and other limitations of the data (e.g. changes in variance across the assay at different antibody titres).

3) Quality of model fitting cannot currently be assessed.

It is difficult to assess how well the model has actually fit the antibody data and thus it's hard to assess how robust the conclusions on different decay half-lives between groups really are. In particular it is unclear whether the authors' assumptions of a single exponential decay applies to all datasets or if the data indicates (for example) a biphasic decay is more appropriate. The authors should:

- i. Include supplementary figures showing the model fits and the data (understanding that lots of data is present, splitting these data into multiple panels for different groups would allow better visualisation)
- ii. The authors should justify that a single exponential decay is appropriate or test whether a model with biphasic decay is more suitable for some groups. This may be particularly relevant where vaccines with high peak titers have a rapid drop in titers, because this might simply be a short first phase of rapid decay followed by a longer half-life of decay?

REVIEWER COMMENTS

Reviewer #1 (Remarks to the Author):

This article uses a large cohort of UK individuals that received multiple vaccine platforms for COVID-19. The authors measure antibody levels after a single or two-dose vaccine in individuals with or without prior SARS-CoV-2 infection. The authors then model antibody decay and perform comparisons. The constricted availability of vaccines make this observation important as it could affect vaccine policy. However, with the emergence of Omicron and other variants it is now more abundant than ever that multiple doses are required for protection and this strategy may not be viable. For example, there is significant Omicron re-infection in individuals with prior infection and significant infection even in vaccinated individuals. Below are specific comments that would strengthen the manuscript.

The major limitation was the timing of the blood sampling was not understood by this reviewer. Then the modeling of antibody decay did not reflect real-world data. First, in figure 1C, levels actually increase over time. Second, in figure 2, older individuals have longer durability. The authors should spend more time describing when the sample was taken and how the modeling of decay is performed.

Response:

We have amended the initial description of the study to make clear that participants were invited to provide initially weekly, then monthly blood samples for antibody testing, which were scheduled independently of any vaccines received. These details were previously provided in the Online Methods, but we agree it is important to provide them in the main text.

Addressing the reviewer's concern that our models may not reflect observed data, we would like to highlight that observed mean antibody levels are shown in Figure S1 with separate panels for each of the main groups studied. The antibody trajectories plotted, where time 0 is first vaccination, show a clear increase in antibody levels after vaccination, followed by a gradual decrease. This pattern forms the basis of our subsequent models and reflects real-world data.

We used 28 days after the first vaccination to represent the peak level for all participants and modelled the antibody decrease from 28 days. In previous Figure 1C, estimates of mean antibody levels increased slightly over time in those who had mRNA-1273 with prior infection, but with substantial uncertainty around the estimates, given the limited number of antibody measurements after 28 days for this group in our previous dataset. Given the ongoing and longitudinal nature of the COVID-19 Infection Survey, we have taken the opportunity to include the most recent data including more participants and antibody measurements, and in our updated Figure 1C, more precise estimates are possible, and these show antibody levels are decreasing, as would be biologically expected.

Regarding older individuals with prior infection having longer durability of antibody responses, this could possibly be due to inherent selection of necessary conditioning on seroconversion for inclusion into the study (without any (initial) detectable response one cannot have a decline/increase). The subset of older participants who seroconverted and were included in the study may have more lasting immune responses than younger participants overall. This is discussed further below.

A second major issue was the correlation to protection equivalence and conjecture about vaccine policy. This is modeled antibody decay, real data would need to be evaluated to inform a vaccine policy change. This would include defining correlates of protection.

Response: In our previous paper (*Wei, J., Pouwels, K.B., Stoesser, N. et al. Antibody responses and correlates of protection in the general population after two doses of the ChAdOx1 or BNT162b2 vaccines. Nat Med (2022)*), we demonstrate that anti-spike antibodies are correlated with protection of infection, and our data in antibody decay are consistent with real-world data of waning vaccine effectiveness. Therefore, we think antibody data can be used to inform vaccine policy as it is strongly correlated with protection.

1. Remove descriptive and personal inferences from abstract and text. I.e “slow and inequitable vaccine roll-out” Are there scientific studies of this to add. Would single doses be faster and more equitable? These statements do not add to the findings. Including the title.

Response: These are factual statements, widely acknowledged by governments, and numerous studies have discussed the issue of a slow and inequitable roll-out of vaccinations in large parts of the world. For example, a recent editorial in the New England Journal of Medicine, with the editor as senior author is entitled “Addressing Vaccine Inequity — Covid-19 Vaccines as a Global Public Good” (N Engl J Med 2022; 386:1176-1179). While we agree such statements may be challenging to read, this provides a context why it may be relevant looking at whether – in light of limited vaccine supplies – using initially only one vaccination among those with previous infection would have generated sufficient levels of protection in a faster way. Without this context there would be little reason to evaluate whether a single dose would have – originally – been sufficient among those with previous infection.

2. Spike binding antibodies do not equate to vaccine protection or efficacy. This needs to be more clear and suggestions of changing vaccine policy due to antibody binding titers that have not been shown to be a correlate of immunity needs to be removed.

Response: As mentioned above, our previous data have shown that binding antibody titres, as measured by the assay used in this current study, are correlated with neutralising activity. Further we have previously shown that higher spike binding antibodies are associated with lower infection risks following both vaccination and natural infection (*Wei, J., Pouwels, K.B., Stoesser, N. et al. Antibody responses and correlates of protection in the general population after two doses of the ChAdOx1 or BNT162b2 vaccines. Nat Med (2022)*). We have added these points to the discussion to make this clearer.

3. There have been studies that examine durability of antibody responses in those with prior infection and vaccination. (eg. Fraley et al. Clin. Infectious diseases 2021.; Zhong et al. JAMA 2021).

Response: These papers examined durability of antibody responses in those with prior infection and vaccination, however these papers examined antibody levels and decline after the second vaccination, while our paper examined antibody response after the first vaccination. We have added these references in our introduction section in page 5 (reference 17-18). While many studies have examined antibody peak levels and half-life post second vaccination, few studies

have examined that post first vaccination. Therefore, our study addresses this gap and provides information on the trajectory of antibody responses post-first vaccination.

4. Comment on individuals that did not respond to first dose. This was over 6,000 people in this study. If you expanded that to global numbers that would be a lot of individuals without any type of immunity if only received a single dose.

Response: We have discussed non-responders to the first dose in our previous publication: *Wei, J., Stoesser, N., Matthews, P.C. et al. Antibody responses to SARS-CoV-2 vaccines in 45,965 adults from the general population of the United Kingdom. Nat Microbiol 6, 1140–1149 (2021).* We have added some discussion and reference to our previous work in page 9-10. Furthermore, among those already having a detectable antibody response due to previous infection, the subgroup potentially eligible for an initial single dose strategy in our paper, all would already have some degree of immunity due to detectable antibody response due to previous infection, and a clear subsequent increase in anti-spike IgG is observed following a subsequent vaccination (e.g. figure 1 from <https://www.nature.com/articles/s41591-022-01721-6>).

5. More clarity on when the antibody levels are measured for each group should be provided.

Response: We have presented the raw antibody data in Figure S1. The X-axis shows the time that antibody was measured relative to first vaccination. We used 28 days from first vaccination to represent antibody peak level and modelled from 28 days. We have clarified this in page 6: “We modelled antibody trajectories using measurements from 28 days post-first dose for all participants (approximate peak levels, Figure S1, S2).”

6. It has been demonstrated with real world data that older individuals have lower durability so there must be an error in the modeling. (Figure 2).

Response:

There are contrasting previous data on the impact of age on antibody declines after infection or vaccination. Reasons for heterogeneity in previous reports include variable definitions of ‘older individuals’, differing durations and timing of follow up, different assays, and different underlying populations amongst others. For example, in a study of healthcare workers increasing age within those of working age (up to 69 years) was associated with longer antibody half-lives post infection (*Clinical Infectious Diseases*, Volume 73, Issue 3, 1 August 2021, Pages e699–e709).

In this current study, we estimated longer antibody half-lives with increasing age (see Table S2). However, as we stress in our discussion, this finding is following conditioning on only studying those who seroconvert. It is only possible to study rates of decline in those who initially respond. We and others have previously reported that seroconversion rates decline with age, so at a population level mean levels after vaccination may be lower at older ages as a result, but these are not necessarily lower in those who seroconvert.

We have added to our discussion to highlight that the slower waning seen in older participants may arise from conditioning on seroconversion.

7. The reference used for correlates of protection levels has not been peer reviewed. The logic of

having lower levels does not track with protection. This results section is pure speculation with modeled data.

Response: The reference used for the correlates of protection levels has now been peer reviewed and published in Nature Medicine. (Wei, J., Pouwels, K.B., Stoesser, N. et al. *Antibody responses and correlates of protection in the general population after two doses of the ChAdOx1 or BNT162b2 vaccines. Nat Med (2022)*).

Reviewer #2 (Remarks to the Author):

Interesting study, for clinicians, policy makers, and the general scientific community.

Only major comments relate to revising the manuscript to make it more concise and to improve the verbiage - see below.

Title would be improved by revising to read: "...vaccine supplies: prior infection and anti-spike..."

The manuscript would benefit from editing to make it more concise, e.g., in abstract, fourth line from the bottom, revise to read: "...threshold was > 1 year after the first dose."

The term "impact" is really not a scientific term. For example, in first paragraph of methods in the main manuscript, the second line would be improved by revising to read: "investigate the relationship between prior infection on..."

The term "at speed" is used at least twice in the manuscript - revise to be more clear.

Response: We have revised the paper accordingly to make it more concise and incorporated the other suggested changes.

Reviewer #3 (Remarks to the Author):

This is a timely manuscript given the increasing prevalence of seropositivity to SARS-CoV-2 in the population, and on-going discussions about vaccine policy and best use of limited supply. The duration findings are particularly relevant.

There are no page or line numbers included in the manuscript, so I will describe the location of questions or proposed edits as best I can below.

Main: Would update section on vaccination rates, and include a web reference rather than older, fixed date references for this informatin, which is in flux.

Response: We have added the reference to WHO COVID dashboard for the vaccination information.

Main.

"and potentially explained by a higher spike protein delivery in MRNA-1273" should be removed or in the discussion - it isn't a result but rather an interpretation of a result.

Response: We have moved this to discussion.

Did you monitor infection after vaccination?

Response: We have data on infection after vaccination. However breakthrough infections after the first vaccination were not the focus of this paper. Furthermore, given the effective roll-out of second vaccinations and relatively low levels of infections during the period where most individuals were between their first and second vaccination, there is not sufficient data to estimate the effect of first vaccination among the subset of survey participants that could be included in the current analysis because of sufficient antibody measurements. For modelling antibodies post first vaccination, we have excluded all measurements after breakthrough infection post-vaccination.

IN the paragraph that beings "we previously" on the 5th line, give the virus strain to which the protection and antibody levels are shown.

Response: We have added in the text that it is Delta variant.

Discussion.

Missing from the discussion is mention of variants of concern, how these might affect results, interpretation and suggestions for programmatic use of number of doses? This seems to be a major deficiency in the current context.

Response: We have briefly discussed the impact of variants of concern in page 12. We mentioned that we need more data to determine the impact of Omicron variant and approaches to vaccine scheduling will need to remain under scrutiny.

Pre-testing at time of first vaccination for antibody seems impractical in most settings - perhaps it could be operationalized in certain high resource settings. How dependent is the antibody testing on training? What is the access to these tests?

Response: Some groups have worked on developing cheap, easy to use COVID-19 antibody tests, that could offer scalable and affordable options for assessing antibody status in LMIC when approved for such use, e.g. <https://www.nature.com/articles/s41467-021-22045-y>; <https://www.nature.com/articles/s41598-021-04298-1>; <https://www.sciencedirect.com/science/article/pii/S0006291X21003971>; <https://journals.asm.org/doi/full/10.1128/JCM.01186-21>; <https://www.mdpi.com/2076-393X/10/3/406/html>. We have added these references in page 10, reference 27-31.

Methods.

Was the decision to exclude non-responders pre-stated, or was that a posthoc decision? How many participants were in the category and how did it influence results?

Response: We decide to exclude non-responders a priori and in line with our previous work. The discussion on vaccine non-responders was in our previous paper (Wei, J., Stoesser, N., Matthews, P.C. et al. *Antibody responses to SARS-CoV-2 vaccines in 45,965 adults from the general population of the United Kingdom. Nat Microbiol* 6, 1140–1149 (2021)), in which we used latent class mixed models to identify a subgroup of non-responders/low responders to the first vaccination (5-6%), who were older, had a higher proportion of males and people with long-term health condition. Therefore, based on that we used a heuristic rule to exclude non-responders from antibody modelling as one cannot determine the peak and decline of non-responders as they simply do not have a detectable response at all. Using this heuristic rule, we identified 4,488 (7%), 1,450 (4%), and 17 (1%) participants as non-responders who received ChAdOx1, BNT162b2, and mRNA-1273, respectively. These are similar to the percentages we previously found using the latent class mixed models. We have added this as discussion in page 9-10.

Reviewer #4 (Remarks to the Author):

The authors present a study that uses antibody binding measurements in a large population level cohort study to determine duration of antibody positivity after vaccination in individuals with and without prior infection. They report that antibody positivity is longer in previously infected individuals who receive one vaccination than in individuals who receive one dose of a vaccine, but not compared to naïve individuals who receive two doses of an mRNA vaccine. The authors argue that this supports a strategy, in vaccine limited contexts, to screen and vaccinate with only one dose those who have had prior infection, but two doses for those without prior infection.

I have a number of major technical and conceptual concerns. In particular, that the primary results presented in this manuscript appear to be also reported in the authors other manuscript referenced as reference 16 in this paper.

Major concern over novelty:

1) This manuscript does not present much that is novel, or perhaps at best presents a very incremental improvement on the authors other paper (reference 16).

The authors here report two main findings, perhaps best summarised in the abstract as:

a. "Prior infection significantly boosted antibody responses for all three vaccines, producing a higher peak level and longer half-life, and a response comparable to those without prior infection receiving two vaccinations."

b. "In those with prior infection, median time above the positivity threshold was estimated to last for >1 year after the first dose."

However, the authors have another manuscript online, which they cite in this paper (reference 16), which reports the exact same conclusions from related data.

- a. "Prior infection significantly increased antibody peak level and half-life with both vaccines."
- b. "protection for vaccinated participants with prior infection, conservatively assuming threshold levels were similar to those vaccinated without prior infection and given their half-lives were longer, the duration of protection could last for >1 year."

I see that the current study has some extra mRNA-1273 data which was not present in the other paper, though this data does not greatly inform the manuscripts overall conclusions. The authors must make clear what is novel in this manuscript over their other manuscripts, but it is my feeling that this work is only an incremental advance on the authors reference 16.

Response: Our previous work (reference 16), now published in Nature Medicine, focused on antibody responses after the second vaccination. The main argument in this paper is antibody responses after the first vaccination and how prior infection influence the results. Therefore, the data used and topics in these two papers are different. We want to examine whether prior infection+single vaccination could provide lasting antibody response and good protection, thus inform vaccine strategies in low-income countries where vaccine supplies are limited. We compared these results with antibody data after two vaccinations in our previous paper (reference 16) to better visualize the difference, but the focus of the current paper is primarily the antibody responses after the first vaccination.

Major technical concerns:

- 2) The estimate of half-life may be confounded by the Upper limit of quantification.

One of the potentially novel aspects of this paper over the authors other paper is the modelling of decay half-lives and peak antibody titers for different groups of individuals. However, this seems confounded by the upper limit of detection of the data on antibody titer used (450 BAU/ml). When fitting a model to estimate the peak level (intercept) and decay rate (slope), the two parameters can easily trade off against each other. That is, sometimes the model will have trouble determining if there was a high peak and rapid decay or a low peak and slow decay. The authors report a number of results where a lower peak was observed in one group but a longer half-life, especially.

"For those with prior infection, the estimated median peak antibody level was 434 BAU/mL (95%CrI 417-454), and the half-life was 333 days (193-1309) at the reference category. The peak levels were lower than previously reported following two BNT162b2 vaccinations without prior infection (974 BAU/mL [942-1009]), but half-life was substantially longer (51 days [50-53])¹⁶."

However, this is worrying because the Upper limit of quantification of the assay is 450. Even though the authors have attempted to deal with the limit of detection using censoring (which is commendable), it is hard to see how the results that the peak is just below the Upper LOD in one group and well above the Upper LOD in another group is going to be a robust result given the LOD is 450. Since the half-lives often trade off with estimated peak level its perhaps unsurprising that one has a long half-life and lower peak, and the other has a high peak and rapid decay. The authors need to demonstrate that their model can robustly estimate the peak level of antibody near or above the upper limit of detection.

The authors would have to do this with a model validation approach, to demonstrate their model is robust to the upper limit of detection and other limitations of the data (e.g. changes in variance across the assay at different antibody titres).

Response: We performed simulation studies to validate that our model is robust to the upper limit of detection. We simulated three scenarios: negative correlation, no correlation, and positive correlation between intercept and slope. Within each scenario, we chose two parameter sets to line up with the different proportion of censored measurements (the proportion of censored data is 11% for ChAdOx1, 19% for BNT162b2, and 38% for mRNA-1273). We used 1000 as the sample size and performed 100 repetitions of each simulation. Our simulation results showed that the Bayesian linear mixed interval-censored model can accurately estimate the intercept and slope in the presence of similar degrees of censoring as observed in the actual data studied, as the model outputs are very similar to the simulated parameters. Therefore, the model we used is robust to the upper limit of detection of the assay and our results are reliable. We have added this point in the limitation section (page 12).

Scenario	Intercept	Slope	Correlation	Proportion of censored (%)	Coverage in intercept	Bias in intercept(95%CI)	Coverage in slope	Bias in slope (95%CI)	Coverage in correlation	Bias in correlation (95%CI)
1	8.5	-0.01	0	13-18	94%	0 (-0.02, 0.02)	94%	0.0001 (-0.0004, 0.0006)	94%	0 (-0.06, 0.08)
2	9.0	-0.01	0	38-44	96%	0 (-0.03, 0.04)	94%	0 (-0.0008, 0.001)	93%	0 (-0.08, 0.1)
3	8.8	-0.02	-0.2	12-16	97%	0 (-0.02, 0.02)	96%	0 (-0.0005, 0.0005)	97%	0 (-0.05, 0.06)
4	9.0	-0.01	-0.5	33-43	95%	0 (-0.03, 0.04)	95%	-0.0001 (-0.0006, 0.001)	98%	0 (-0.06, 0.05)
5	8.8	-0.02	0.5	17-22	93%	0 (-0.02, 0.03)	95%	0 (-0.0004, 0.0005)	96%	-0.01 (-0.05, 0.04)
6	9.0	-0.01	0.2	39-45	94%	0 (-0.02, 0.03)	96%	-0.0001 (-0.0006, 0.0006)	95%	0.01 (-0.06, 0.05)

3) Quality of model fitting cannot currently be assessed.

It is difficult to assess how well the model has actually fit the antibody data and thus it's hard to assess how robust the conclusions on different decay half-lives between groups really are. In particular it is unclear whether the authors' assumptions of a single exponential decay applies to all datasets or if the data indicates (for example) a biphasic decay is more appropriate. The authors should:

- i. Include supplementary figures showing the model fits and the data (understanding that lots of data is present, splitting these data into multiple panels for different groups would allow better visualisation)
- ii. The authors should justify that a single exponential decay is appropriate or test whether a model with biphasic decay is more suitable for some groups. This may be particularly relevant where vaccines with high peak titers have a rapid drop in titers, because this might simply be a short first

phase of rapid decay followed by a longer half-life of decay?

Response: To examine non-linearity in antibody decline, we fitted a model using four-knot splines for time (knots places at 10th, 33rd, 67th, and 90th of observed time points) and compared the model fit (using LOOIC) and estimated trajectories with those from the linear model. The model fit is worse (LOOIC higher) for the spline model compared to the linear model for ChAdOx1 (155331 vs 138883) and mRNA-1273 (7586 vs 7570). The model fit for spline model is slightly better than the linear model for BNT162b2 (107858 vs 108863). However, as shown in the plot below, trajectories from linear and spline models were similar for all three vaccine groups, and there was no evidence of antibody decline flattening to any meaningful extent up to 77 days post-first vaccination. Therefore, we used the linear exponential model to examine antibody decay, and the effects from different covariates. We have added the plot as Figure S4 and the comparison of model fit in the Methods.

REVIEWER COMMENTS

Reviewer #1 (Remarks to the Author):

The authors chose not to adequately address many of the concerns and in turn referenced their recent manuscript in many of the comments. Thus, it is unclear what the advance this manuscript has over the previous article. Many of the policy and correlations to protection comments and conjecture are better suited for a review or editorial article and not a scientific paper reporting results. Could be placed in the discussion, but not in the main results and title as this is misleading. Simply measuring anti-spike antibody levels and modeling decay would not be utilized to support vaccination policy.

Reviewer #4 (Remarks to the Author):

The authors have addressed both my technical concerns very clearly with additional analyses and I have no further comments regarding this. However, I remain unconvinced that the results of this work stand particularly separate from the authors' Nature Medicine publication and appear incremental over the previous work, but this is a matter of opinion.

In particular, although the authors argue that the work in Nature Med focused on two doses (instead of one dose here), they still previously reported results based from the single dose data, and it is unclear that the data are indeed different here. Instead it seems that the analysis here focuses on a largely overlapping subset of what was included in the Nature Medicine paper. What is and is not overlapping data, analysis or results as the authors' previous paper should be highlighted more clearly.

I.e. the authors' main conclusion here (i.e. one of two conclusions highlighted in the abstract):

"Prior infection significantly boosted antibody responses for all three vaccines, producing a higher peak level and longer half-life, and a response comparable to those without prior infection receiving two vaccinations."

Is very similar to the result reported in the Nature Medicine paper:

"For ChAdOx1, participants without prior infection had lower antibody levels after the second dose than those with prior infection after the first dose; but, for BNT162b2, two vaccinations without prior infection led to higher antibody levels than previously infected participants having only one dose, especially for 80-year-old participants (Fig. 1; trajectories for other dosing intervals in Extended Data Fig. 4)."

Which seems to arise from analysing overlapping data?

Further, the conclusion about the predictions of the duration of immunity in the Nature Medicine paper read very similarly to those presented in this manuscript. The authors have highlighted in their response, the difference of this calculation from the second dose in the Nature Medicine paper versus the first dose here, but this was not obvious on first read, especially because the result of >1 year is the same for both.

Here:

"Data were insufficient to estimate correlates of protection for those with prior infection¹⁶, but, since levels associated with the same degree of protection were lower for unvaccinated individuals, if we conservatively assume the threshold levels are similar post any vaccination, the duration providing >67% protection is estimated to be around 170-220 days for a single ChAdOX1 vaccination, and over a year for a single BNT162b2 vaccination in those with prior infection."

Nature Medicine:

"Although data were insufficient to estimate antibody levels correlated with protection for vaccinated participants with prior infection, conservatively assuming that the threshold levels were

similar to those vaccinated without prior infection, and given that their half-lives were longer, the duration of protection could last for more than 1 year. Times for antibody levels to fall to the threshold for positivity—that is, 23 BAU ml⁻¹—were longer but followed the same patterns (Extended Data Fig. 7).”

The authors could make this similar result more clear across the two studies. Perhaps this could be done by including at lines 159-164 a comparison between the 2 dose (with prior infection – from Nature Med) with the 1 dose (with prior infection – from this paper). Currently this section compares results to the Nature Medicine paper, but only for those with no prior infection who had two doses. If I understand correctly, this additional comparison could improve clarity and highlight that two doses is not much better than 1 dose in those with prior infection?

REVIEWER COMMENTS

Reviewer #1

1. The authors chose not to adequately address many of the concerns and in turn referenced their recent manuscript in many of the comments. Thus, it is unclear what the advance this manuscript has over the previous article. Many of the policy and correlations to protection comments and conjecture are better suited for a review or editorial article and not a scientific paper reporting results. Could be placed in the discussion, but not in the main results and title as this is misleading. Simply measuring anti-spike antibody levels and modeling decay would not be utilized to support vaccination policy.

The most important and substantive differences regarding first vaccinations – the focus of the current paper – between this manuscript and the previous Nature Medicine paper are as follows:

	Current paper	Nature Medicine paper
Antibody trajectories after first dose	Data from 8 December 2020 to 31 January 2022, including substantially more first	Data from 8 December 2020 to 4 October 2021

	vaccinations and follow-up time in younger individuals, who dominate in LMIC Data on mRNA-1273 included Fully adjusted Bayesian linear mixed interval-censored models – accounting for upper limit of quantification of the assay - were used to estimate changes in antibody levels after the first vaccination. Furthermore, models allowed for different antibody trajectories, and hence duration of protection, through including interactions in the fully adjusted model.	NO data on mRNA-1273 First dose only analysed descriptively using minimally adjusted (only for age and dosing interval) generalised additive models that do not account for the upper limit of quantification of the assay and could not be used to estimate half-lives, predict beyond the observed data or evaluate responses in specific groups.
Prediction of duration of protection after first dose only	Included	Not included, only focused on predicted duration of protection after 2 vaccinations.
NEW (based on reviewer 4): comparison of protection of 1dose+infection vs 2dose+infection	Comparison of protection of duration after 1 dose + infection vs 2 doses + infection	Only focused on predicted duration of protection after 2 vaccinations, no data on first vaccination
Estimation of the impact of a vaccination prioritisation strategy based on LFIA antibody tests as a measure of previous infection	Included, including a Shiny app which facilitates evaluating the impact of other assumptions about costs, sensitivity, and specificity of a test	Not included (also not possible using the analyses and included data)

We have clarified these substantive differences in the revised text, highlighting what information is coming from the Nature Medicine paper, and have incorporated the additional comparison suggested by reviewer 4. Furthermore, we have toned down the potential implications for policy based on these data, as suggested.

In addition, we have changed the title to “SARS-CoV-2 antibody trajectories after a single COVID-19 vaccination with and without prior infection”.

Reviewer #4

1. The authors have addressed both my technical concerns very clearly with addition analyses and I have no further comments regarding this. However, I remain unconvinced that the results of this work stand particularly separate from the authors Nature Medicine publication and appear incremental over the previous work, but this is a matter of opinion.

In particular, although the authors argue that the work in Nature Med focused on two doses (instead of one dose here), they still previously reported results based from the single dose data, and it is unclear that the data are indeed different here. Instead it seems that the analysis here focuses on a largely overlapping subset of what was included in the Nature Medicine paper. What is and is not overlapping data, analysis or results as the authors previous paper should be highlighted more clearly.

I.e. the authors main conclusion here (i.e. one of two conclusions highlighted in the abstract):

“Prior infection significantly boosted antibody responses for all three vaccines, producing a higher peak level and longer half-life, and a response comparable to those without prior infection receiving two vaccinations.”

Is very similar to the result reported in the Nature Medicine paper:

“For ChAdOx1, participants without prior infection had lower antibody levels after the second dose than those with prior infection after the first dose; but, for BNT162b2, two vaccinations without prior infection led to higher antibody levels than previously infected participants having only one dose, especially for 80-year-old participants (Fig. 1; trajectories for other dosing intervals in Extended Data Fig. 4).”

Which seems to arise from analysing overlapping data?

Further, the conclusion about the predictions of the duration of immunity in the Nature Medicine paper read very similarly to those presented in this manuscript. The authors have highlighted in their response, the difference of this calculation from the second dose in the Nature Medicine paper versus the first dose here, but this was not obvious on first read, especially because the result of >1 year is the same for both.

Here:

“Data were insufficient to estimate correlates of protection for those with prior infection¹⁶, but, since levels associated with the same degree of protection were lower for unvaccinated individuals, if we conservatively assume the threshold levels are similar post any vaccination, the duration providing >67% protection is estimated to be around 170-220 days for a single ChAdOX1 vaccination, and over a year for a single BNT162b2 vaccination in those with prior infection.”

Nature Medicine:

“Although data were insufficient to estimate antibody levels correlated with protection for vaccinated participants with prior infection, conservatively assuming that the threshold levels were similar to those vaccinated without prior infection, and given that their half-lives were longer, the duration of protection could last for more than 1 year. Times for antibody levels to fall to the threshold for positivity—that is, 23 BAU ml⁻¹—were longer but followed the same patterns (Extended Data Fig. 7).”

The authors could make this similar result more clear across the two studies. Perhaps this could be done by including at lines 159-164 a comparison between the 2 dose (with prior infection – from Nature Med) with the 1 dose (with prior infection – from this paper).

Currently this section compares results to the Nature Medicine paper, but only for those with no prior infection who had two doses. If I understand correctly, this additional comparison could improve clarity and highlight that two doses is not much better than 1 dose in those with prior infection?

Please see the substantive differences between this and previous work summarised in the table above – in particular we have added the additional analysis suggested by the reviewer. We have also ensured that we have clearly signalled where any estimates have come or comparisons are made with the previous paper (now highlighted in yellow, but already incorporated in the previous version, already including references each time).